# The Gut Microbiome as a Catalyst and Emerging Therapeutic Target for Parkinson’s Disease: A Comprehensive Update

**DOI:** 10.3390/biomedicines12081738

**Published:** 2024-08-02

**Authors:** Rebecca Kerstens, Paul Joyce

**Affiliations:** Centre for Pharmaceutical Innovation (CPI), UniSA Clinical & Health Sciences, University of South Australia, Adelaide, SA 5000, Australia; kerry009@mymail.unisa.edu.au

**Keywords:** microbiota, probiotics, prebiotics, microbiome therapeutics, microbial therapies, personalised medicine

## Abstract

Parkinson’s Disease is the second most prevalent neurological disorder globally, and its cause is still largely unknown. Likewise, there is no cure, and existing treatments do little more than subdue symptoms before becoming ineffective. It is increasingly important to understand the factors contributing to Parkinson’s Disease aetiology so that new and more effective pharmacotherapies can be established. In recent years, there has been an emergence of research linking gut dysbiosis to Parkinson’s Disease via the gut–brain axis. Advancements in microbial profiling have led to characterisation of a Parkinson’s-specific microbial signature, where novel treatments that leverage and correct gut dysbiosis are beginning to emerge for the safe and effective treatment of Parkinson’s Disease. Preliminary clinical studies investigating microbiome-targeted therapeutics for Parkinson’s Disease have revealed promising outcomes, and as such, the aim of this review is to provide a timely and comprehensive update of the most recent advances in this field. Faecal microbiota transplantation has emerged as a novel and potential frontrunner for microbial-based therapies due to their efficacy in alleviating Parkinson’s Disease symptomology through modulation of the gut–brain axis. However, more rigorous clinical investigation, along with technological advancements in diagnostic and in vitro testing tools, are critically required to facilitate the widespread clinical translation of microbiome-targeting Parkinson’s Disease therapeutics.

## 1. Introduction

Parkinson’s Disease (PD) is the second most prevalent neurological disorder, affecting approximately 3% of people aged over 65 globally [1]. One of the greatest risk factors for PD is age, and with the global trend towards aging populations, the number of PD sufferers is expected to double by 2030 [2]. Only 5–10% of PD is thought to be familial PD, which is heritable and caused by genetic mutations [3]. Sporadic (or idiopathic) parkinsonism is the most common form of PD that manifests as progressive protein aggregation through the formation of misfolded α-synuclein clusters, known as Lewy bodies, which accumulate in the neurons of the central nervous system (CNS), as well as the enteric nervous system (ENS) of the gastrointestinal tract (GIT) [4]. Some environmental toxins, such as certain pollutants and pesticides (such as rotenone and paraquat) or drugs (MPTP, a toxic by-product of the opioid analgesic desmethylprodine, MPPP), have been identified as agents that can cause PD [5]; however, for the vast majority of those affected by PD, the catalyst is unknown [6].

PD is characterised by motor dysfunction, such as tremor, rigidity, postural instability, and bradykinesia [7,8]. These hallmarks of PD are caused by damage to the substantia nigra pars compacta (SNc). The motor symptoms start to appear when a majority of the dopaminergic neurons are already damaged in the SNc [9], and approximately 80% of dopamine levels are depleted in the nigrostriatal terminals [10]. Unfortunately, this means PD diagnosis and treatment are delayed until substantial neurological damage has already occurred, resulting in suboptimal treatment and reduced quality of life in PD sufferers [11].

In addition to motor dysfunction, patients experience non-motor symptoms, often including cognitive or behavioural abnormalities, anxiety, depression, sleep disturbances, hyposmia/anosmia, and gastrointestinal (GI) symptoms, such as constipation and delayed gastric emptying [1,8,12]. Importantly, evidence shows that GI symptoms, especially constipation and intestinal inflammation, occur up to decades before motor symptoms [13,14]. As these symptoms are experienced by virtually all PD patients (gastric motility disorders: 70–100%, constipation: up to 90% of PD patients) [13], these symptoms are important indicators of early-stage PD and may even be implicated in the pathogenesis of the disease. The pathogenesis of PD is still poorly understood, and existing treatments for PD are inadequate, often only alleviating symptoms for a few years before becoming ineffective or causing side effects, such as dyskinesia [15]. There is currently no cure for Parkinson’s, nor any way to slow its progression. Thus, understanding PD aetiology and developing new pharmacological treatments for the disease is vital.

Recently, there has been an emergence of research into the bi-directional interaction between the brain and gut microbiome, known as the “gut–brain axis” [16]. The gut–brain axis is a complex communication network mediated by the CNS, the ENS, and the various microbe populations that inhabit the gut microbiome. Communication is maintained through the vagus nerve, as well as the immune system, tryptophan metabolism, and microbial metabolites, such as short-chain fatty acids, branched-chain amino acids, and peptidoglycans [16,17]. As research into this area expands, evidence points to the role of the gut–brain axis in neurological diseases, including PD [18]. This review updates recent findings that outline the link between gut microbiome disturbances (commonly referred to as gut dysbiosis) and PD development, progression, and severity, providing clear rationale for the development of novel microbiome-targeted therapies for the effective prevention and treatment of PD. Here, we summarise recent microbiome-targeted PD therapies in light of the role of gut dysbiosis in PD, building on other quality reviews on this topic [10,14,16,19,20,21,22]. It is envisioned that insights derived from this timely updated review will provide future directions for researchers to engineer new therapies that enable improved management of this debilitating disease with few current effective treatment options.

## 2. The Gut Microbiome as a Key Driver of Parkinson’s Disease

In 1913, Dr. Friederich Lewy put forth the notion of the vagus nerve as a centre of brain pathology [4]. By 2003, Professor Heiko Braak elaborated on this in the context of PD by presenting the hypothesis that an unknown pathogen in the gut is the culprit behind sporadic PD. Braak’s “dual-hit theory” suggests that this pathogen may enter the gastrointestinal tract (GIT) from the nasal passage and affects neurons in the gut and nasal cavity [6]. This pathogen leads to α-synuclein protein aggregation in the GIT, which then travels to the CNS and higher cortical regions via the vagal nerve and the olfactory tract. Subsequent studies have shown that α-synuclein deposits can be detected in GIT biopsies prior to the onset of PD, and post-mortem examinations have shown similar accumulation of these proteins inside the olfactory bulb of the CNS [23]. The pioneering work performed by Lewy and Braak has served as the foundation for the wealth of subsequent studies that explore the role of the gut in PD, with emerging evidence linking the gut microbiome with disease progression and severity. Aside from the vagal communication method highlighted by Braak’s theory, the gut microbiome may also communicate with the central nervous system in other ways, including hormonal and immune signalling. For a detailed outline of the other mechanisms, the reader is referred to Cryan et al., 2019 [17]. The purpose of this section was to review the evidence that links the gut microbiome with PD, providing an overview for the microbial signatures of PD and the mechanisms by which gut dysbiosis triggers neuroinflammation and degeneration, ultimately highlighting the rationale for gut microbiome-targeted therapies for treating PD.

## 3. Gut Dysbiosis as a Microbial Signature of PD

The human gut microbiome is composed of more than 100 trillion microorganisms, including bacteria, yeasts, and viruses, along with their genomes and metabolome [22]. The gut microbiome differs between individuals, and is affected by diet, antibiotics, environment, physical activity, and age [18,23]. Early life also impacts microbial composition, with birth delivery type, infant feeding practices (i.e., breast milk versus formula), and gestational age playing important roles [21,24]. In humans, *Bacillota* (formerly *Firmicutes*), *Bacteroidota* (formerly *Bacteroidetes*), *Actinomycetota* (formerly *Actinobacteria*), *Pseudomonadota* (formerly *Proteobacteria*), *Fusobacteria*, and *Verrucomicrobia* are the main phyla found in the GIT. *Bacillota* (formerly *Firmicutes*) and *Bacteroidota* (formerly *Bacteroidetes*) are typically dominant, contributing to 90% of the microbial population [22]. A healthy gut microbiome is integral to host wellbeing, with important roles in maintaining intestinal barrier integrity, function, and metabolism, as well as immune system function and regulation of the gut–brain axis [21]. Additionally, gut microbes play an important role in digestion, degrading indigestible dietary fibres via fermentation, and producing short-chain fatty acids (SCFAs) and other metabolites that are essential for host health [23].

While there is still no consensus on what the ideal microbiome composition is in a healthy human, there are distinct changes in the gut microbial colonies of PD sufferers. For example, small intestinal bacterial overgrowth (SIBO) is present in 25% of PD patients, a prevalence that is significantly higher than that in healthy adults [9]. Since technological advancements in genome sequencing have made the classification of the various microbial populations in the human gut more accurate and accessible, there has been a push to identify characteristic compositional shifts as ‘microbial signatures’ of numerous diseases, including PD. Several studies have analysed human faecal samples to highlight the key changes in taxa at the phyla, family, and genera levels, where the findings indeed indicate that there are specific microbial shifts that characterise the PD gut microbiome (Figure 1). Evidence is emerging that would link various changes in abundance of key taxa at the phyla, family, and genera levels to the onset, progression, and severity of PD. These data are summarised in Table 1, which outlines shifts in the relative abundance of microbes by taxa in PD patients compared with healthy controls. However, it is noted that many studies do not consider the impacts of age and sex on microbial composition, whether in patients or healthy controls. While differences exist in microbiota composition in PD patients across each study, there are consistent results across key genera: numerous studies show an increase in the relative abundance of *Akkermansia* and *Bifidobacterium*, and decreases in *Blautia*, *Faecalibacterium*, and *Roseburia* in PD faecal samples.

As highlighted in Table 1, it is not possible to pinpoint any one responsible bacterium or microbial taxa in the pathogenesis of PD, since it is not necessarily the presence or the absence of certain microbes, but rather overall compositional shifts, known as dysbiosis, that contribute to PD. Dysbiosis refers to alterations in the number and composition of gut microbes, the gut microbiome environment, and microbial metabolites [21]. These alterations in microbial communities are frequently measured and quantified via alpha (i.e., microbial richness) and beta diversity (i.e., variations between microbiota within two or more hosts). In their systematic review of gut microbiome studies between PD patients and healthy controls, Boertien et al. [35] found that alpha diversity indices were reported in eleven of thirteen assessed studies; of these, three found higher alpha diversity in PD, six reported no difference, one reported lower alpha diversity, and one reported a difference without suggesting whether it was increased or decreased, further highlighting the inter-study variability in microbial signatures of PD. However, the authors noted that in all studies that performed 16S rRNA gene sequencing or metagenomic sequencing, beta diversity showed differences between PD patients and healthy controls, indicating changes are present [35].

## 4. Key Microbial Alterations in PD at the Genus Taxonomical Level

### 4.1. Akkermansia

Akkermansia are some of the most important bacteria under the Verrucomicrobiaceae family, which degrade mucous for energy. While these mucin-degrading bacteria are vital for intestinal barrier homeostasis, their over-abundance leads to degradation of the intestinal mucous barrier and contributes to increased intestinal permeability [8], which may expose the intestinal neural plexus directly to oxidative stress or toxins, such as lipopolysaccharide (LPS) and pesticides [20,36]. This can lead to aggregation of α-synuclein fibrils and the subsequent generation of Lewy bodies and neuroinflammation in the ENS [14,37]. Numerous studies have shown that increased abundance of Akkermansia increases the risk of PD, as well as accelerating disease progression [37,38].

### 4.2. Bifidobacterium

While the literature shows conflicting information about Bifidobacteria, when considering statistically significant results from Table 1, PD patients appeared to have a relative increase in abundance of this genus, compared with healthy controls. Interestingly, Bifidobacteria are typically considered commensal bacteria, and may be protective against PD symptoms. For example, Dogra et al. [9] suggested that low counts of Bifidobacteria are associated with worsening of hallucinations. The main role of Bifidobacteria, and other bacteria in the Bifidobacteriaceae family, is to prevent the overgrowth of harmful bacteria and regulate the GI immune system. Bifidobacterium is also an important producer of the SCFA acetate [37]. Thus, it is suggested that relative over-abundances may indicate potential over-compensation in an attempt to reconstruct gut homeostasis [24].

### 4.3. Blautia

Blautia, and other relevant genera implicated in PD pathology, such as Coprococcus, Dorea, Lachnospira, Roseburia, and Ruminococcus, fall under the Lachnospiraceae family. This family is important for its role in hydrolysing diet-derived polysaccharides to produce butyrate and other SCFAs [39]. As shown in Table 1, bacteria from the Lachnospiraceae family are often depleted in PD patients, and numerous studies have found Blautia to be consistently decreased in PD. SCFAs, such as butyrate, modulate the activity of the ENS and thereby increase GI motility. Therefore, the reduction in Blautia, and other SCFA-producing bacteria, likely leads to the gastrointestinal dysmotility exhibited in PD [40]. Additionally, butyrate can suppress colon inflammation [39], which is another symptom implicated in PD pathology.

### 4.4. Coprococcus

Bacteria in the Coprococcus genus are key SCFA-producers under the Bacillota (formerly Firmicutes) phylum. These Gram-positive, obligate anaerobic cocci ferment carbohydrates to produce butyric and acetic acids. The relative abundance of this genus is associated with a reduced risk of many neuropsychological and neurodegenerative disorders. Conversely, the species *Coprococcus eutactus*, of the Coprococcus genus, is depleted in adults with PD [41]. Coprococcus and other genera from the Lachnospiraceae family, such as Blautia and Roseburia, are associated with anti-inflammatory properties [21]. It is noted that one study reviewed in Table 1 showed conflicting evidence: Li et al. [27] found that Coprococcus was increased rather than decreased, which necessitates further studies to elucidate the role of Coprococcus in PD.

### 4.5. Faecalibacterium

Faecalibacterium facilitate the degradation of cellulose and starch, fermenting these indigestible fibres into SCFAs, including butyrate [13,21]. The resultant butyrate and other anti-inflammatory metabolites support GI health, so the reduction in Faecalibacterium may impair gut barrier function. This can increase the risk of pathogenic invasion and α-synuclein formation in the ENS [24]. Additionally, Faecalibacterium, similar to other SCFA-producing bacteria, have been associated with anti-inflammatory properties [42]. SCFAs have also been shown to protect against dopamine and tyrosine hydroxylase depletion in the SNc [42]. Decreases in Faecalibacterium abundance are noted in PD faecal samples, and this change is likely accelerated with disease progression [43]. Furthermore, Faecalibacterium is also decreased in both Chron’s disease and ulcerative colitis; thus, the reduction in PD patients may cause similar symptoms to inflammatory bowel diseases, which could then lead to PD pathology [38].

### 4.6. Roseburia

Roseburia is another important butyrate producer [13] that plays an important role in strengthening the intestinal barrier, as butyrate supports the maintenance of tight junctions and mucin production by enterocytes. This is important for preventing microorganisms from crossing into the lamina propria [13,44]. Reductions in SCFA-producing bacteria may lead to increased intestinal permeability, also known as “leaky gut” [9]. Additionally, butyrate exhibits anti-inflammatory effects via the induction of regulatory T cells and downregulation of pro-inflammatory cytokines and Toll-like receptor (TLR) 4 receptors [39]. Some species in the Roseburia genus also produce acetate. This SCFA is beneficial in inhibiting the growth of entero-pathogens, reducing luminal pH, and increasing the absorption of dietary nutrients [39].

As well as the consistent depletion of Roseburia evidenced in PD patient faecal samples, studies have shown significant correlations between lower abundance of Roseburia and worse clinical progression of motor and non-motor symptoms [37]. According to Vacca et al. [39], Blautia and Roseburia are the genera most involved in the control of gut inflammatory processes, which is mediated through their metabolism of butyrate.

## 5. Factors Contributing to Inconsistent and Variable Microbial Signatures of PD

Despite the consistent changes in key taxa highlighted above, the findings summarised in Table 1 also demonstrated distinct inconsistencies for changes in the relative abundance of other key taxa in PD patients. For example, Petrov et al. [33] reported a decrease in *Bacteroides* in PD patients, while Keshavarzian et al. [25] reported an increase, with both findings being statistically significant (in faecal samples). The inconsistencies may be due to a multitude of factors, including the following:***Stage of disease:*** Multiple studies have demonstrated that the gut microbiome composition varies considerably for various stages of PD patients [1,9,22]. For example, one study [25] took samples from both medication-naive PD subjects as well as treated PD patients of various disease durations. They noticed significant differences in microbiota composition between patients of different disease stages and highlighted that the significance was maintained when the medication-naive samples were omitted from the analysis. Thus, they concluded that while medications may have an effect, disease duration significantly impacts microbial communities in PD patients.***Patient variations, including ethnicity, age, and dietary habits:*** Some patient variables are well-known determinants of microbiome composition, for example, age and sex [35]. There are also differences found between ethnicities, for example, bacteria from the *Lactobacillaceae* family are typically enriched in Western PD cohorts, but the same has not been found in Chinese PD studies [45]. Aside from racial differences, this variability between populations could be due to geographic location, and related dietary and/or environmental factors [27]. For instance, one study, which took samples from patients in three locations across the United States of America, identified geographical location as a confounding factor [26]. Dietary habits are also an important confounder, as the types of food consumed are directly related to the survival of various microbial populations. Of particular relevance is fruit and vegetable consumption, as these foods supply the dietary fibres that many microbes use as energy.***Sampling method:*** Sample collection, transport, and storage, laboratory procedures, and sequencing methods can all impact upon the microbial composition of samples. For example, the “gold standard” for transport and storage of stool samples is to freeze samples at −80 °C immediately upon collection, which will preserve the samples for up to two years. Alternatively, samples can be stored at −20 °C for a few months [35]. Of the 11 studies reviewed in Table 1, only 5 mentioned freezing samples. Two of these [29,32] immediately froze stool samples to −80 °C and −35 °C, respectively. A further two stated that samples were collected at home before being frozen (at −80 °C and −20 °C, respectively) but provided no information on the time of collection and freezing [13,31]. Finally, Li et al. [27] noted that subjects were advised to store their own samples at −20 °C in a freezer until collection (within three days) when the samples were frozen at −80 °C. However, it is unlikely that subjects were able to adhere to this protocol in their home freezer. Thus, storage and transportation conditions can be a potential source for both intra- and inter-study differences [35]. Likewise, there are composition differences between faecal and mucosal samplings. For example, there are environmental differences between the colonic mucosa and the lumen, which lead to distinct differences in biodiversity and bacterial taxa [37]. Most studies characterise the microbiome through faecal extraction and microbial sequencing. However, one utilised both faecal and mucosal sampling in their study. They found that as well as the expected differences between microbiota collected from PD patients compared with healthy subjects, significant differences in α-diversity were observed between mucosal and faecal microbiota within both groups [25]. Finally, differences in sample DNA extraction methods and subsequent sequencing methods are also contributing factors to the differences found between these studies.***Other contributing factors, including medication usage:*** While it can be difficult to extricate the effect of medications from the impact of disease duration on the gut microbiome composition of PD patients, it is apparent that medications and pharmaceutical formulations can affect microbiota composition [46,47]. For example, catechol-o-methyltransferase (COMT) inhibitors and anticholinergics led to significant differences in gut microbiome composition in PD patients [26]. Furthermore, both classes of drug led to GI side effects, which can further contribute to gut dysbiosis. While Hill-Burns et al. [26] only found a borderline significant effect of levodopa on gut microbiome composition, some strains of *Lactobacillus* produce enzymes that degrade levodopa into dopamine. This implies that levodopa use in PD patients may increase the abundance of certain *Lactobacillus* strains [45].

Ultimately, the inconsistencies and variations in microbial profiles of PD patients provide further evidence for the complexity of microbiome analyses and pinpointing a microbial signature for disease states, such as PD, considering the multifaceted factors that contribute to microbiome ecology, which introduces challenges in the design and implementation of gut microbiota-targeted therapies for widespread use across all PD patients.

## 6. Mechanisms Linking Gut Dysbiosis with PD Pathology

The altered abundance of various microbes is linked to PD pathology through a multitude of mechanisms, since all microbial genera/families exert different roles within the GIT. For example, an increase in pathogenic bacteria can lead to the production of toxins, tissue damage, and inflammation, while the relative increase in typically commensal bacteria can crowd other bacteria, dysregulate homeostatic functions, and/or alter the metabolome. In this way, dysbiosis refers to both the changes in relative abundance of microbes, at different taxonomical levels, and in the microbiome environment (including mucin layers, epithelial barriers, TLR, etc.) and metabolome. These factors all interact to induce and accelerate disease pathology via various factors, schematically summarised in Figure 2. Specifically, the mechanisms that link gut dysbiosis to PD pathology are detailed in Table 2, which provides key examples of compositional changes in the gut microbiome that are known to induce such mechanisms in PD patients. The focus of this review was too ambitious to provide a complete understanding of all the mechanistic actions involved in the pathogenesis of PD, which have been covered extensively in prior literature [8,9,16,20,22,24] (Dogra et al., 2022; Huang et al., 2021; Liang et al., 2021; Lorente-Picón and Laguna, 2021; Rani and Mondal, 2021; Shen et al., 2021; Wang et al., 2021). Thus, these key mechanisms and outcomes of dysbiosis have been highlighted, as they can potentially be corrected by microbiome-targeting therapies.

## 7. Emerging Gut Microbiome-Targeted Therapies for Treating Parkinson’s Disease

As evidence continues to emerge linking gut dysbiosis with idiopathic PD, burgeoning research also highlights potential therapies that can effectively and safely mitigate symptoms and slow PD progression by targeting the gut microbiome. These include antibiotics, probiotics, prebiotics, postbiotics, and faecal microbiota transplantation (or faecal matter transplantation, FMT; Figure 3). A summary of the research and most recent evidence in the field of novel PD treatments follows.

### 7.1. Antibiotics

Antibiotics, as well as clearing pathogenic microbes, can have other biological actions in the CNS, such as anti-inflammatory, immunomodulatory, and antioxidant effects, while also demonstrating potential to prevent abnormal protein aggregation [20]. Additionally, the repurposing of existing drugs (e.g., conventional antibiotics) can be beneficial, as safety profiling has already been completed and they are approved for use in humans, which accelerates the process of approval for clinical trials and their use for treating a new indication (e.g., PD). Numerous studies, including clinical trials and animal models, have examined the effects of various antibiotics on neurological disorders. Current evidence shows that antibiotics may exert neuroprotection by modulating gut microbiota and ameliorating PD-like pathophysiology [16]. Table 3 highlights some of the key antibiotic drugs that have demonstrated positive indications in PD models.

In addition to those listed in Table 3, combinations of antibiotics may also be useful to correct microbial imbalances. Additionally, there are established antibiotic treatments for *H. pylori* infection and SIBO, which have been linked with PD [37].

While some antibiotics show promise in helping to rectify microbial imbalances, the use of antibiotics as a therapy should be approached with an abundance of caution. Antibiotics can lead to dysbiosis through the disappearance of commensal bacteria, and the proliferation of new bacterial species. Thus, it is important to understand the mechanisms of various antibiotics, and their interactions between not only pathogenic bacteria, but also the wider microbiome population, ensuring that the beneficial bacteria are preserved and protected [20]. For example, minocycline showed promise as a potential drug candidate to treat PD and transitioned into a Phase II clinical trial in 2006 and 2008. However, the results of these trials did not show beneficial results and highlighted that long-term use of minocycline may be of concern [20]. Thus, more investigations are required to demonstrate their clinical potential in treating or alleviating PD symptomology.

### 7.2. Probiotics

The International Scientific Association for Probiotics and Prebiotics defines probiotics as “live microorganisms that when administered in adequate amounts confer a health benefit on the host” [51]. Recent studies using various probiotics have shown promising results in patients with PD. For example, administration of fermented milk containing *Lactobacillus casei Shirota* has shown therapeutic potential towards the GI symptoms of PD [52]. Indeed, probiotic use, in conjunction with prebiotics, has been shown to alleviate constipation [53]. Likewise, beneficial effects have been shown using probiotic strains of Lactobacillus and Bifidobacterium [16]. Probiotics may also be used during the management of *Heliobacter Pylori* infections to help restore the balance of commensal bacteria after antibiotic treatment. In PD patients, this helps to improve the absorption of levodopa [37].

In a 12-week randomised, double-blind, placebo-controlled clinical trial, PD patients who received a probiotic (containing *Lactobacillus acidophilus*, *Bifidobacterium bifidum*, *Lactobacillus reuteri*, and *Lactobacillus fermentum*) had better scores on the Movement Disorders Society–Unified Parkinson’s Disease Rating Scale (MDS-UPDRS), compared to the control group [54]. Reductions in oxidative stress biomarkers (such as C-reactive protein and malondialdehyde) and increased glutathione were also recorded. In other studies, *L. salivarius LS01* and *L. acidophilus LA02* significantly decreased oxidative stress and pro-inflammatory cytokines and promoted the production of anti-inflammatory cytokines in PD patients [16]. This may be because probiotics can upregulate the production of anti-inflammatory vitamins (for example, vitamins E, D3, B6, riboflavin, biotin, cobalamin, folates, pantothenic acid, nicotinic acid, and pyridoxin) and other bioactive molecules [37]. Finally, there is an indication that most probiotic strains also influence GI membrane integrity and inhibit potentially pathogenic bacterial overgrowth [16].

Animal models of PD have also shown that probiotic treatment improves PD outcomes. For example, *L. plantarum DP189* has been shown to reduce α-synuclein aggravation by reducing oxidative damage, inflammation, and gut microbiota dysfunction [14]. Likewise, oral probiotic administration over a period of 16 weeks had neuroprotective effects in a transgenic PD mouse model, via the preservation of dopaminergic neurons [50]. The progressive deterioration of motor functions was also arrested: quantitative assessments showed that the treatment mitigated balance, coordination, and gait impairments. Probiotics also regulate the microbiome and impact upon serotonin metabolism, and other microbiome-targeting interventions can modulate the function of neurotransmitters, including dopamine and serotonin. However, further studies are needed to understand the impact of probiotics on neurotransmitters in PD patients.

While studies have produced exciting results, the use of probiotics for the treatment of illnesses is contentious. This is due in part to issues surrounding the processing and storage of probiotic supplements that use live bacteria. For example, changes in nomenclature and the lack of regulation in this space make it difficult to determine what probiotic products contain [55]. Additionally, there can be issues with whether storage materials (including capsules and jars) and storage environments (for example, temperature) are conducive to maintaining live bacteria [56]. Further to storage challenges, evidence suggests that traditional probiotics rarely survive the passage through the acidic environment of the stomach and/or effectively colonise the GIT without being excreted. In fact, recent evidence suggests that rather than live bacteria, bacterial spores are not only more likely to survive the stomach, but they are also more stable during processing and storage [41]. Alternatively, an emerging technique that can help to maintain the survivability of probiotics through processing and storage, as well as consumption and transit through the GIT, is through microencapsulation, which refers to incorporating probiotics into biopolymer- or emulsion-based delivery systems. The reader is referred to the review by Yoha et al. [57], which discusses the importance and various methods for encapsulation of probiotics. Encapsulation can also improve the stability and adhesion of probiotics, via the use of different pH, enzymatic, or immune responsive systems, which allow for targeted delivery of probiotics to specific areas within the GIT [58].

Finally, if the bacteria do make it alive to the intestines, sufficient fuel sources (e.g., prebiotic fibres) are needed to support their subsequent colonisation of the GIT. In this regard, the use of a combination of probiotics with prebiotic fibres, known as synbiotics, has shown to support the survival of probiotics [16]. Another important consideration about the use of probiotics as a treatment is that some strains of bacteria that are typically seen as beneficial or commensal can lead to SIBO or disease if they become over-abundant. For example, *Akkermansia*, increasingly implicated as a catalyst of PD, is often utilised in probiotics to improve host health. Thus, supplementation without knowledge of patients’ gut microbiome composition could potentially cause undue harm.

### 7.3. Prebiotics

Prebiotics are beneficial to human health, as they stimulate the growth and activity of health-promoting bacteria, in particular favouring the growth of SCFA-producing bacteria [37,39]. Prebiotics are indigestible dietary components, including fructo-oligosaccharides (FOS), inulin, starches, lactulose, galacto-oligosaccharides (GOS), etc., which are fermented by gut bacteria in the colon for energy [20,39,41]. This fermentation results in the production of beneficial SCFAs as by-products. As SCFAs and other by-products of fermentation are typically acidic, prebiotics can modify the gut via changes in pH. For example, pH alteration can change the population of acid-sensitive species, such as *Bacteroides*, and promote butyrate formation by *Bacillota* (formerly *Firmicutes*), in a process known as the butyrogenic effect [56]. Prebiotic-related increases in SCFA reduce intestinal barrier permeability, inhibit endotoxin crossing the mucosal barrier, and reduce inflammation. Prebiotic fibres are also beneficial in promoting GI motility, stool quality, immune function, and alleviating constipation [20,22]. Animal studies have also indicated that supplementation with prebiotics can increase the levels of brain-derived neurotrophic factor (BDNF) in the dentate gyrus of the hippocampus. These effects result in decreased neuroinflammation and reduced dopaminergic neuron loss [37].

While supplementation with prebiotic fibres has positive results, simple dietary changes are also an effective way to increase prebiotic consumption. For example, the Western diet, which consists of limited dietary fibre and increased amounts of animal protein, sugars, and saturated fatty acids, is increasingly implicated in chronic disease, including PD. A recent systematic review found that Western dietary patterns not only increased the incidence of PD, but also exacerbated disease severity in patients [14]. This is because the Western diet leads to a reduction in carbohydrate-fermenting bacteria, and thus, reduced butyrate, propionate, and acetate [20,22]. Conversely, numerous studies have concluded that the Mediterranean dietary pattern, with its high consumption of vegetables, fruits, nuts, and fish, and thus fibre, polyphenols, and omega-3 fatty acids, lowers the risk of PD, and may exert neuroprotective effects in PD patients [14,41]. This may be, in part, due to the higher total SCFAs found in those with greater plant-food consumption. Increased consumption of omega-3 fatty acids, vitamins, and polyphenols in this dietary pattern is also a contributing factor. Clinical studies have shown that supplementation with omega-3 in conjunction with vitamin E has beneficial effects in PD patients [59,60]. Supplementation with omega-3 also supports the proliferation of SCFA-producing bacteria from the *Lachnospiraceae* family (such as *Blautia* and *Coprococcus* genera, and *Roseburia* spp. species) [39]. By limiting ultra-processed foods, and increasing consumption of high-fibre foods, which include fruits and vegetables, nuts, legumes and beans, and wholegrain carbohydrates, prebiotic intake is naturally increased. The increase in dietary prebiotics and other beneficial nutrients will support the proliferation of commensal, SCFA-producing bacteria.

Although prebiotic fibres are typically found in a balanced diet, chitosan oligosaccharide (COS) is emerging as another promising prebiotic. Unlike other prebiotics, COS is not a dietary fibre—it is obtained from the shells of certain crustaceans. COS has been shown to reduce oxidative stress and oxidative damage, while exerting an anti-inflammatory response by reducing the release of pro-inflammatory cytokines [53]. Furthermore, COS is beneficial when utilised for probiotic encapsulation, and it has strong adhesion properties, which support the colonisation of probiotics in the colon [36,57]. Despite the promising results emerging in recent research for the use of prebiotics across a range of disease states linked with gut dysbiosis, limited evidence exists for their use as a PD pharmacotherapy. A recent study assessed the impact of different types of prebiotic fibres on SCFA production in PD patients compared with healthy controls. The authors concluded that all fibre types stimulated SCFA production. However, they observed that butyrate production remained lower in PD patients than healthy humans, regardless of prebiotic supplementation [61]. Thus, while prebiotics may exert beneficial results in PD patients, they are best treated as an adjunct therapy to support microbiome health by providing energy to commensal bacteria and improving GI function.

### 7.4. Postbiotics

SCFAs and other metabolites, including bioactive peptides, tryptophan degradation metabolites, bile acids, and vitamins (such as vitamins B1, B3, B9, B12, K, and A), that are produced by probiotic microbes are considered postbiotics and may be supplemented without the need for administering the live bacteria [62]. The importance of SCFAs and other by-products of microbial fermentation cannot be overstated. Covered in detail previously, SCFAs have several functions that impact host wellbeing and homeostasis. Some of these important impacts include maintaining the integrity of the intestinal barrier, promoting normal microglial development, anti-inflammatory effects, and mitigating neuronal damage [37]. In recent years, postbiotic supplementation has been explored due to increasing evidence showing the immunomodulatory, anti-inflammatory, anti-proliferative, and antioxidant effects of SCFAs. Studies have shown that butyrate administration alleviates motor symptoms, improves dopamine deficiency, and reduces neuroinflammation in PD animal models [14]. Animal models have also shown that oral supplementation of butyrate increases plasma concentrations, indicating that oral administration may have direct actions in the brain [20].

A recent study investigated the effects of sodium butyrate treatment on PD-induced mice [63]. This study utilised ceftriaxone as a pre-treatment to induce dysbiosis and gut alterations, followed by a 6-hydroxydopamine interstitial injection to catalyse dopaminergic damage in order to mimic Braak’s ‘dual-hit theory’ of PD in the mice. The mice were subsequently treated with sodium butyrate. This study showed that butyrate treatment repaired gut damage by significantly increasing goblet cell numbers and Muc2 expression in the intestines of the mice, reducing intestinal inflammation and gut permeability. Treatment also led to improvements in motor coordination and neuro-behavioural deficits, which the authors associated with reduced pathogenic factors and inflammation in the striatum, as well as reduced systemic inflammation. The authors concluded that butyrate treatment led to improvements in microbiome composition, helping to establish a new microbial balance with positive effects on the PD phenotype.

Tryptophan is also metabolised by gut microbiota, as well as via the kynurenine pathway or the serotonin pathway [64]. The subsequent metabolites produced can be either neuroprotective (such as kynurenic acid (KYNA), picolinic acid, and nicotinamide adenine dinucleotide), or neurotoxic (such as quinolinic acid (QA) and 3-hydroxykynurenine (3-HK)). In PD patients’ brains, the balance is shifted in favour of neurotoxic tryptophan metabolites: KYNA is decreased in the putamen, SNc, and the frontal cortex, while QA in plasma and 3-HK in the SNc and putamen are increased. The accumulation of these metabolites increases neurotoxicity and oxidative stress. Melatonin, which is another metabolite of the serotonin pathway, decreases oxidative stress and the expression of mitochondrial-dependent apoptotic pathways. This prevents the destruction of dopaminergic neurons and relieves the non-motor symptoms of PD [64]. A recent meta-analysis of randomised controlled trials utilising melatonin supplementation in PD patients suggested that melatonin supplementation reduces motor symptoms and sleep disturbances in patients [65]. However, the five included studies each had small sample sizes, with the meta-analysis only including a total of 155 patients. Furthermore, the authors suggested that more promising results might be observed if melatonin was administered in early-stage PD patients.

Despite indications that postbiotic supplementation could offer hope to PD patients, there have been few studies that utilise postbiotic supplementation in humans. Furthermore, while supplementation with postbiotics is an exciting area of research in the treatment of neurological disorders, it is noted that altering the microbial composition of the gut microbiome would lead to increased microbially derived metabolites. Microbially produced metabolites act both locally and systemically via the gut–brain axis. Thus, treatments that work to shift a patient’s microbiome, from a PD microbial signature to that of a healthy human, would likely be a more effective long-term solution.

### 7.5. Faecal Microbiota Transplantation

FMT uses stool from a healthy donor to recolonise the gut microbiome in a patient. Faecal matter is transferred into the patient via colonoscopy, nasal-jejunal tube, or orally via capsules [36,66]. FMT is perhaps the most advanced microbiome-targeted therapy for a wide range of disorders linked with gut dysbiosis, in large part due to its holistic approach, whereby native microbiomes (including microbes, fibre, and metabolites) are transplanted into the patient’s GIT. As such, FMT is already an established treatment for *Clostridium difficile* infection and IBD and has shown promising results in patients with various neurological disorders [2,20,40,67,68]. FMT is also associated with beneficial changes in microbial composition, such as increased relative abundance of *Blautia* (Genus) and *Lachnospiraceae* (family), and significantly reduced abundance of *Escherichia-Shigella* (Genus) [37]. The use of FMT in PD patients is still under-researched; however, multiple clinical trials (and one case study) have revealed promising results, as outlined in Table 4.

The most statistically powered FMT clinical trial in PD patients to date is the recent study by Cheng et al. [70], which included 54 patients (27 received treatment and 27 a placebo). In this study, improvements were seen in roughly half of the treatment group, termed “FMT responders” (FMT.R). The responders showed statistically significant improvements in MDS-UPDRS total scores and non-motor symptoms, including GI symptoms. Interestingly, the FMT treatment participants showed improvements in cognitive function, regardless of their responses in PD symptoms. The authors noted that a high placebo response was observed with regard to GI symptoms at weeks 4 and 8. They suggest that this may be due to dietary recommendations given to study participants. Regardless, by week 12, the FMT arm showed significant improvements, which were not matched in the placebo group.

To understand why some patients responded to FMT while others did not, the study utilised metagenomic sequencing to analyse microbial species in the FMT.R and FMT non-responder (FMT.NR) subgroups. At baseline, there were no significant differences in microbiota diversity between either subgroup and taxa were found with significantly different abundance. By the 12th week, there was still no significant microbiota diversity between the two subgroups, although 20 microbial species had notably altered levels in FMT.R individuals compared with FMT.NR. Furthermore, some of the altered species had strong correlations with improvements in patient clinical scores. The authors highlighted *Eubacterium eligens*, *Eubacterium ventriosum*, *Clostridiales bacterium 42_27*, *uncultured Blautia* sp., *Clostridioides difficile*, *uncultured Clostridium* sp., and *Roseburia hominis* as key species that were positively correlated with GI and PD symptoms. Additionally, they found that the two subgroups had different traits in gut microbial functional pathways, which also positively correlated with the improvement in both GI and PD symptoms. Unfortunately, the follow-up time of this trial was short, at just 12 weeks. Future studies with longer observation periods and more detailed supporting dietary interventions will complement these findings.

Beyond Cheng et al.’s study, Table 4 highlights the capacity for FMT to alleviate gastric symptoms, such as constipation, in PD patients, with some studies observing positive responses in motor symptoms. However, these studies are not without limitations. Firstly, the sample sizes are very small, with 4 of the 7 studies having sample sizes of between 6 and 15 subjects. Additionally, three of the seven clinical trials did not analyse patients’ microbiome composition before or after FMT through longitudinal studies. Furthermore, in most of the studies, it does not appear as though patients were encouraged to eat diets that would support the colonisation of the new microbiota, post-treatment. Where this recommendation was made, the implementation of dietary measures was lacking. Kuai et al. [40] suggested that “all the patients basically followed the traditional Chinese Food structure (containing mainly grains and vegetables, small amounts of meat) before and after the FMT treatment”. However, no further mention of dietary methods is made in the article. Likewise, Cheng et al. [70] suggested that participants were advised to follow a “specific regular and healthy diet, with limited protein intake and high fibre”; however, the authors also noted that no dietary information was recorded.

In animal models of PD, FMT has been evidenced to increase dopamine levels in the SN, reduce neuroinflammation, modulate immune responses, decrease α-synuclein expression, and reduce motor symptoms [20,67]. Animal studies have also shown that FMT is likely also neuroprotective in PD by inhibiting TLR4-NFκB pathway-mediated inflammation [21]. Furthermore, Sun et al. [71] showed that mice receiving gut microbiota from PD mice also displayed motor dysfunctions, as well as decreased striatal dopamine, serotonin, and their metabolites. Conversely, the PD mice receiving faecal microbiota from normal mice displayed significant recovery of dopamine, serotonin, and their metabolites in striatum, as well as recovery of motor function. They also highlighted that FMT can suppress the loss of dopaminergic neurons in the SN in animal models.

## 8. Current Knowledge Gaps and Future Directions

Recent research has built upon the early works of Lewy and Braak, to highlight the involvement of the gut–brain axis in the pathogenesis and progression of PD. Exciting developments have been made when it comes to determining the ‘PD microbial signature’, as well as leveraging the gut microbiome in treatments for PD. However, more research is vital. One of the biggest limitations to progress is the absence of defined research protocols. Likewise, differences in sampling methods make it difficult to compare results between studies. This affects both studies that determine the composition of the gut microbiome and those that leverage the microbiome for treatment. Another area for improvement is the recruitment of patients and controls for clinical trials. Where clinical trials have shown promise, often a small sample size limited the statistical power of the results. Thus, there is a need for larger studies with high participant numbers to accurately determine the PD microbial signature at various stages of PD, as well as to demonstrate efficacy for promising treatments, such as FMT. A recent conference abstract referred to a study that sought to characterise the PD gut microbiome in a large cohort, including 1009 PD patients and 546 neurologically healthy controls [72]. Unfortunately, at the time of writing, it does not appear as though these data have been published. The results from this analysis, as well as other ongoing studies, will add value to the current knowledge base, and serve as precedent for more clinical trials in the future. Other knowledge gaps and promising future directions are summarised below.

### 8.1. Microbiome Composition and Non-Motor Symptoms as Diagnostic Markers

As important as determining the PD microbial signature and researching novel treatments for the disease is identifying new diagnostic tools to detect PD and potentially mitigate some of the neurological damage. As it stands, there are currently no reliable biomarkers to predict the onset or early stages of PD. Often the pathway to diagnosis involves medicating patients with motor symptoms to see if they respond to dopamine replacement therapy. However, this review has highlighted numerous symptoms and risk factors of PD that occur decades before the onset of motor symptoms. These include common non-motor symptoms, such as constipation, rapid eye movement (REM) sleep behaviour disorder, excessive daytime sleepiness, and postprandial fullness [73]. Anosmia and taste loss, mood disturbances, excessive sweating, fatigue, and pain can also occur up to ten years prior to motor symptoms. Patients who experience these symptoms should be considered for biological testing, which may indicate potential PD biomarkers.

PD patients often exhibit reduced production of thiamine and folate, which presents as deficiency [16]. Riboflavin metabolism, which is reflective of riboflavin status, is also reduced, and vitamin D deficiency is also common [73,74]. Nutrient levels, and the presence of lipopolysaccharide-binding protein (as a marker for LPS presence), can be determined via serum testing. Likewise, total and active levels of plasma ghrelin are decreased in patients with PD, which can also be determined by a blood test [22]. The presence of *H. pylori* and SIBO have also been identified as risk factors for PD, both of which are measurable by non-invasive breath tests (carbon and lactulose hydrogen breath tests, respectively). Other microbiome-related PD indications include gut inflammation, reduced SCFAs, GI presence of α-synuclein proteins and LPS, and microbial signatures. For example, reduced abundance of *Prevotellaceae* in faeces of PD was considered a biomarker of PD [13]. Similarly, reductions in SCFA producers, such as *Ruminococcus*, *Blautia*, and *Faecalibacterium*, might be an indicator of PD. Furthermore, reduced faecal SCFAs and higher faecal calprotectin levels also correlate with PD [49]. Additionally, increased pathobiont species, such as *Enterococcus* and *Escherichia-Shigella* can also be indicative of PD [13]. Thus, faecal sampling should be utilised in patients at risk of PD to ascertain the presence of a ‘PD microbial signature’ and identify the presence of calprotectin and other inflammatory markers.

Taken individually, each of the above factors are likely not strong enough to predict PD. However, together, they could form a diagnostic tool that may be utilised in the identification of early-stage PD, or perhaps even to identify and mitigate risk prior to disease onset. For example, a simple symptom checklist could serve as an indication for combined biological testing, including blood and breath testing, and faecal sampling. The development of such a clinical tool will lead to improved and earlier diagnoses of PD, and thus enable quicker and more effective treatment. Likewise, improved diagnostic tools will mitigate the risk of unnecessarily medicating patients with similar presentations.

### 8.2. Technological Advancements Required for the Development of Next-Generation Diagnostic Tools and Pharmacotherapies

Considering the obvious role of the gut microbiome in the pathogenesis of PD, defining the PD microbial signature for use as a diagnostic tool is pertinent. Data from previous studies can be utilised to determine epidemiological associations between specific microbial taxa and PD. The influence of individual genetics on the abundance of specific taxa is also an important consideration. This should be established in genome-wide association studies (GWAS) [35]. Furthermore, analysis of microbial metabolites and their functions is essential to comprehend the relationship between the gut microbiome and human health, and in doing so it is important to differentiate microbial metabolites from the host or food-derived components [75]. Thus, the use of machine learning (ML) and other new technologies will be vital in the development of diagnostic tools. Emerging technologies in this space are already showing promise, as outlined below.

***Machine Learning:*** Recently, computational approaches have led to greater understanding of the relationship between gut microbes, human genetics, and microbial metabolites, as integration of various ‘omics’ methods (for example, metataxonomic and metagenomic sequencing, and metabolomics) with machine learning is becoming increasingly common. For example, the gutMGene database has linked 332 gut microbes, 223 genes, and 207 microbial metabolites in humans. Furthermore, the relationship between 774 gut microbiota and 221 human diseases has been elucidated in the Amadis database, and the relationship between 579 gut microbiota and 77 intervention measures or 123 disorders in humans according to the gutMDisorder database [75].

In 2020, Pietrucci et al. [34] utilised a machine learning approach to determine microbial shifts in PD patients. They re-analysed data from studies comprising a total of 472 PD patients and 374 controls, randomly selecting 80% of the samples to create the training set, and the remaining 20% to create the test set and evaluate the prediction. They compared efficacy between three different ML algorithms and found that the RF algorithm was the most effective, with an accuracy of 71%. Using RF, they found a subset of 22 bacterial families that were predictive of PD and characterised them by importance. The authors noted that the importance of each family in PD pathology was not directly correlated with its relative abundance in the gut microbiome. For example, similar to the previous studies, the authors found that *Verrucomicrobiaceae/Akkermansiaceae* and *Bifidobacteriaceae* are key bacterial taxa associated strongly with PD pathogenesis; however, they noted that these species have a low relative abundance. Interestingly, not all the families identified as important by the algorithm have been previously identified in the literature. For example, two of the ten most important bacterial families, *Veillonellaceae* and *Alcaligenaceae*, have never been reported before. These families were both relatively more abundant in PD patient samples.

This work identified ML as a powerful tool to determine the PD microbial signature for diagnostic purposes; however, more work needs to be carried out. The data samples used were a small fraction of the available information, and a larger dataset may help to improve accuracy. Furthermore, inconsistencies in methodologies used in studies may also impact the results. Identification of a standard protocol for microbiome studies and data sharing between researchers will be key to effectively utilising ML in PD microbiome research. Advances in the genetic characterisation of microbes will enable researchers to identify the key species of bacteria (and their metabolites) involved in PD. This will support the development of precision interventions to target the specific bacteria involved in PD aetiology and progression, as opposed to current microbiome-targeting therapies, which are indiscriminate in their approach.

**In Vitro *models:*** Developing biologically relevant in vitro models that simulate the gut microbiome and gut–brain axis and benefit from strong in vitro–in vivo correlations is critical for the testing and optimisation of next-generation PD therapeutics. Human intestinal organoids (HIO) are cellular models of the human intestine established through differentiation of induced pluripotent stem cells. These models can then be ‘reprogrammed’ using patient-derived materials, to create minimally invasive, patient-specific assessments. Patient ENS tissue and gut microbiota sampling can be used to determine significant contributions of gut microbiota to the pathophysiology of PD. These models can also be used to determine the efficacy of novel therapeutic targets that target the microbiome.

Another novel in vitro model is the Mucosal-Simulator of the Human Intestinal Microbial Ecosystem (M-SHIME^®^), which is a simulator of the human intestinal microbial ecosystem, equipped with a mucosal compartment. Ghyselinck et al. [76] used M-SHIME^®^ to determine the efficacy of probiotic supplementation in restoring bacterial composition in PD patients. While their study was small (6 subjects: 3 PD and 3 control), and although the data covered only a 48 h period, some significant changes in bacterial composition and microbial metabolites were observed. While more studies using these models are needed, HIOs, M-SHIME^®^, and other new in vitro models will be valuable tools in the development and testing of novel treatments for PD.

### 8.3. Dietary Interventions to Support Microbiome-Targeted Therapies

Diet is an important consideration when designing microbiome-targeting interventions, since it is well established that many commensal bacteria rely on indigestible dietary fibres for energy sources. While dietary interventions are not recommended as a treatment for PD, ensuring accurate measurement and recording of dietary consumption prior to, and following, microbiome-altering treatments (such as FMT and probiotic supplementation) would strengthen the methodological reporting of these interventions. Moreover, adequate dietary fibre consumption can prevent mucin degradation, as some bacteria will break down and utilise intestinal mucous for energy in the absence of prebiotic fibres. Thus, ensuring that any introduced bacteria have adequate ‘food’ sources will support their subsequent colonisation of the GIT, and prevent damage to the GI lining. Furthermore, healthy diets that incorporate adequate dietary fibre can also help to alleviate some of the GI symptoms involved in PD.

Aside from dietary fibres, other dietary components and nutrients have benefits in supporting the proliferation of commensal bacteria. For example, a pilot human clinical trial confirmed that the relative abundance of *Coprococcus* spp. increased upon administration of vitamins A, B2, C, and D3. Furthermore, the SCFA concentration also increased according to the type and dose of administered vitamins. Another example is coffee, which is widely regarded as a negative risk factor towards PD. Coffee may also be neuroprotective, likely due to a combination of the caffeine and polyphenols. Coffee consumption can impact the gut microbiome, leading to an increase in anti-inflammatory *Bifidobacteria* and decreased pathogenic bacteria, such as *Clostridium* spp. and *Escherichia coli*. In PD mouse models, coffee regulates the gut microbiota, reducing α-synuclein aggregation and mitigating dopaminergic neuronal loss, as well as improving motor symptoms [14].

***Improved understanding of the role of non-bacterial microbes in the human gut microbiome***: Despite the surge of interest in the human gut microbiome over recent decades, up to 65% of bacterial species residing in the human GIT remain uncultured, so understanding of their biological roles is lacking [64]. Moreover, most of the literature revolves around the bacterial microbiome, and there is a lack of understanding of the role of the fungal, viral, and eukaryotic microbiomes [17]. There is also a lack of tools for sequencing fungal and viral genomes, or analysing their functions, as well as limited reference databases in this space [77]. Determining the role of viruses in the GIT is vital to understanding the role of the gut microbiome in diseases, including PD, since viruses exert a direct impact on the human host, as well as the other residing microbes within the microbiota. While fungi make up a smaller portion of the gut microbiome compared to viruses, studying their impact on the microbiome is also important, as there appears to be a correlation between increased fungal communities and gut dysbiosis, as seen in IBD [17].

## 9. Conclusions

It is now clear that the gut microbiome has a role in both the aetiology and treatment of PD. Studies characterising the PD microbial signature have come a long way in recent years to identify the specific microbial imbalances that characterise gut dysbiosis in PD. As more trials are completed, researchers will better understand the complete PD microbial signature, as well as early indications of the disease. With these advancements, there is hope that soon we will be able to detect PD at a much earlier stage, or even identify high-risk individuals before the onset of disease. This will potentially mitigate the severe neurological damage seen in the later stages of PD. Importantly, results from both preclinical and clinical studies have shown that the gut microbiome can be leveraged to alleviate PD symptoms via advanced microbiome-targeting therapies. However, more extensive clinical testing and development is required to optimise microbiome-targeting therapies for the treatment of PD, with specific focus on adequately controlled and high-powered clinical studies that validate treatment efficacy. Additionally, further studies must be completed to determine at which stage of disease microbiome-targeted therapies may be deployed for beneficial results. With future technological advancements in diagnostic tools, in vitro testing, and patient-centric formulation strategies, it is expected that microbiome-targeted therapies may serve as pharmacotherapies for improving the quality of life of PD sufferers, either as standalone treatment or in conjunction with current care practices and future advancements

## Figures and Tables

**Figure 1 biomedicines-12-01738-f001:**
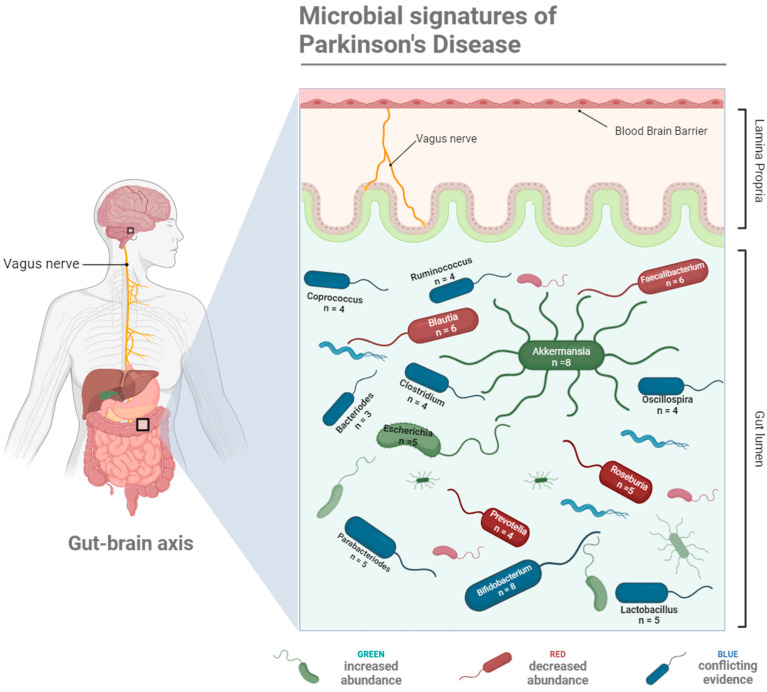
A schematic representation of the key microbial taxa linked with Parkinson’s Disease onset, progression, and severity via the gut–brain axis, serving as microbial signatures of Parkinson’s Disease. Taxa coloured green have been observed to increase in abundance in PD patients, taxa coloured red decrease in abundance in PD patients, and conflicting evidence relating to their relative abundance in PD patients exists for taxa coloured blue. The number of studies providing evidence for each microbial shift are provided in the labels. Created with BioRender.

**Figure 2 biomedicines-12-01738-f002:**
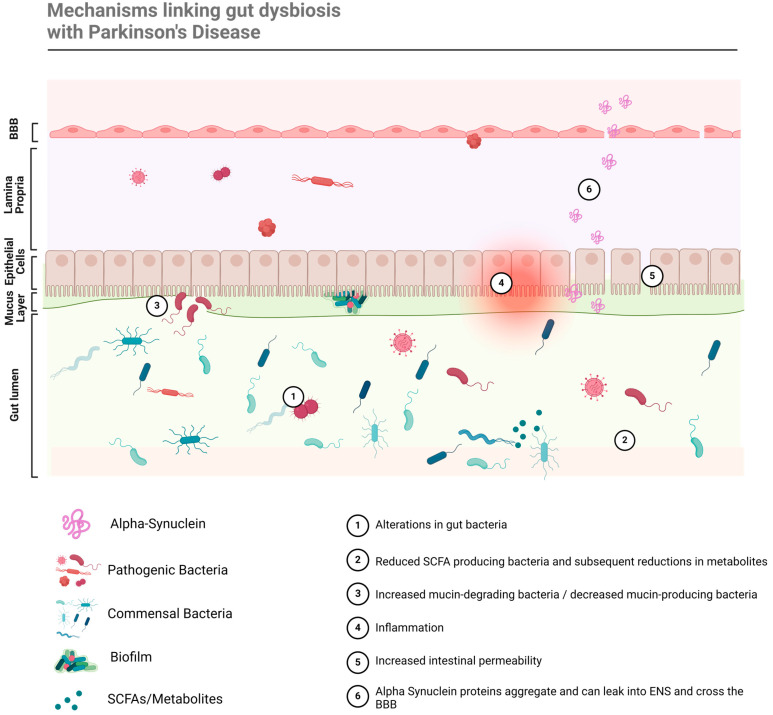
Gut dysbiosis is linked with PD pathogenesis through a multitude of mechanisms that lead to neuroinflammation and neurodegeneration.

**Figure 3 biomedicines-12-01738-f003:**
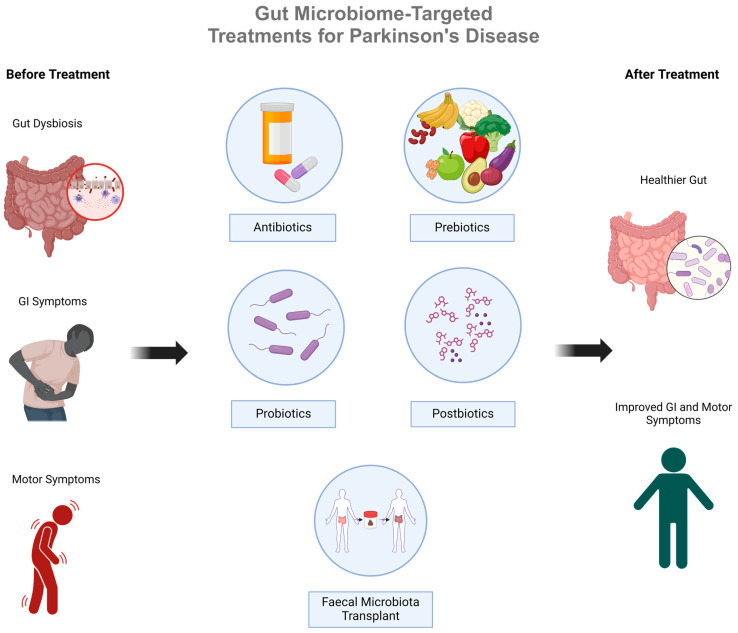
An overview of gut microbiome-targeted therapies for Parkinson’s Disease that aim to overcome gut dysbiosis by restoring the gut microbiome in PD patients.

**Table 1 biomedicines-12-01738-t001:** Variations in human gut microbiome composition, grouped by taxa, between healthy controls and Parkinson’s Disease patients, based on faecal matter and mucosal sample analysis [4,13,25,26,27,28,29,30,31,32].

	Bedarf et al., 2017 (faecal) [4]	Keshavarzian et al., 2015 (mucosal) [25]	Keshavarzian et al., 2015 (faecal) [25]	Hill-Burns et al. 2017 (faecal) [26]	Li et al., 2017 (faecal) [13]	Li et al., 2019 (faecal) [27]	Petrov et al., 2017 (faecal) [33]	Pietrucci et al., 2019 (faecal) [34]	Unger et al., 2016 (faecal) [29]	Vascellari et al., 2020 (faecal) [30]	Wallen et al., 2020 (faecal) [31]	Zhang et al., 2020 (faecal) [32]
Phylum												
*Actinomycetota* (formerly *Actinobacteria*)		↓	↓		↑*					↑*		↑*
*Bacteroidota* (formerly *Bacteroidetes*)		↑	↑*		↓*				↓*	↓*		↓*
*Cyanobacteria*										↓*		
*Bacillota* (formerly *Firmicutes*)		↓	↓*		N/C				N/C			↑*
*Fusobacteria*												↓*
*Pseudomonadota* (formerly *Proteobacteria*)		↑	↑*		↑*					↑*		↑*
*Verrumicrobia*		↑	↑*			↑*				↑*		↑*
Family												
*Alcaligenaceae*										↓*		
*Coprobacillaceae*		↓*	↓*									
*Bacteroidaceae*		↑	↑*							↓*		
*Bifidobacteriaceae*		↓	↓	↑*						↑*		↑*
*Brevibacteriaceae*										↓*		
*Clostridia*		↑	↓									
*Clostridiaceae*		↑	↑*									
*Comamonadaceae*										↓*		
*Coriobacteriaceae*		↑	↑							↑*		
*Christensenellaceae*				↑*								
*Desulfovibrionaceae*										↑*		↑*
*Enterobacteriaceae*		↑	↑					↑*	↑*			
*Enterococcaceae*								↑*				
*Erysipelotrichaceae*	↓*	↓	↓									
*Fusobacteriaceae*												↓*
*Lachnospiraceae*		↓	↓*	↓*				↓*		↓*		↑*
*Lactobacillacae*				↑*		↓*		↑*				
*Microbacteriaceae*										↑*		
*Oxalobacteraceae*		↑*	NC									
*Pasteurellaceae*				↓*								
*Peptostreptococcaceae*		↓	↓									
*Porphyromonadaceae*		↑	↑			↑*						↑*
*Prevotellaceae*	↑*	↓	↑									
*Pseudomonadaceae*		↓	↓									
*Rikenellaceae*		↓	↑			↑*				↑*		
*Ruminococcaceae*		↓	↑			↑*						↑*
*Sphingobacteriaceae*										↓*		
*Streptococcaceae*										↑*		
*[Tissierellaceae]*				↑*								
*Veillonellaceae*		↓	↑			↑*						↑*
*Verrucomicrobiaceae*	↑*	↑	↑*	↑*		↑*						↑*
Genus												
*Acetobacterium*										↑*		
*Acidaminococcus*											↑*	
*Actinomyces*											↑*	
*Akkermansia*	↑*	↑	↑*	↑*		↑*			↑	↑*		↑*
*Alistipes*						↑*						
*Anaerostripes*										↑*	↓*	
*Bacteroides*		↑	↑*				↓*					
*Bifidobacterium*		↓	↓	↑*			↑*		↑*	↑*	↑*	↑*
*Blautia*		↓	↓*	↓*	↓*					↓*	↓*	
*Brevibacterium*										↓*		
*Butyricicoccus*											↓*	
*Butyrivibrio*										↓*		
*Catabacter*							↑*					
*Christensenella*							↑*				↑*	
*Caldicellulosiruptor*										↑*		
*Cloacibacillus*											↑*	
*Clostridium*		↑	↓							↑*	↓*	
*Coprobacillus*											↑*	
*Coprococcus*		↓	↓*			↑*				↓*		
*Corynebacterium*											↑*	
*Desulfonauticus*										↑*		
*Desulfovibrio*										↑*		
*Dolichospermum*										↓*		
*Dorea*							↓*					
*Eisenbergiella*											↑*	
*Enorma*											↑*	
*Enterobacter*										↑*		
*Enterococcus*					↑*						↑*	
*Erysipelatoclostridium*											↑*	
*Escherichia*		↑	↑							↑*	↑*	↑*
*Escherichia-Shigella*					↑*							
*Eubacterium*	↓*	↓	↓								↓*	
*Faecalibacterium*		↓*		↓*	↓*		↓*		↓*		↓*	
*Fusicatenibacter*											↓*	
*Fusobacterium*												↓*
*Harryflintia*											↑*	
*Hungatella*											↑*	
*Klebsiella*										↓*	↑*	
*Lactobacillus*				↑*		↓*	↑*		↓*		↑*	
*Lacnospira*										↓*		
*Megasphaera*											↑*	
*Methanobrevibacter*						↑*			↑		↑*	
*Methanomassiliicoccus*											↑*	
*Monoglobus*											↓*	
*Odoribacter*										↓*		
*Oscillospira*		↓	↑*				↑*					↑*
*Parabacteriodes*		↑	↑			↑*				↓*		↑*
*Peptostreptococcus*		↓	↓									
*Phascolarctobacterium*						↑*						↑*
*Porphyromonas*											↑*	
*Prevotella*	↓*	↓					↓*		↓			
*Prosthecobacter*										↑*		
*Pseudobutyrivibrio*										↓*		
*Ralstonia*		↑*	↓									
*Roseburia*		↓	↓*	↓*						↓*	↓*	
*Ruminococcus*		↓	↓		↓*	↑*					↓*	
*Ruthenibacterium*											↑*	
*Scardovia*											↑*	
*Serratia*										↑*		
*Slackia*										↑*		
*Sutterella*										↓*		
*Streptococcus*					↑*	↓*				↑*		
*Turicibacter*											↑*	
*Veillonella*										↑*		

N/C = no change in abundance; ↑ = increase in abundance; ↓ = decrease in abundance; ↑* = statistically significant increase in abundance; ↓* = statistically significant decrease in abundance.

**Table 2 biomedicines-12-01738-t002:** A summary of the various outcomes of gut dysbiosis, caused by specific composition changes in the gut microbiome, that are linked with Parkinson’s Disease pathogenesis.

Mechanism	Compositional Changes to the Microbiome that Trigger Defined Mechanisms *	Impact of Bacterial Alterations	Link to PD Pathogenesis
Intestinal barrier damage	*Akkermansia* ↑ *Prevotellaceae* (*family*) ↓	*Akkermansia* is a mucous-degrading Gram-negative bacteria that digests intestinal mucous for use as an energy source. *Prevotellaceae* are a family of Gram-negative bacteria that synthesise mucin. An imbalance of *Akkermansia* and/or *Prevotellaceae* leads to excess mucin degradation and reduced intestinal barrier integrity.	Permeability of the intestinal mucous barrier is increased, triggering a pro-inflammatory response in the intestines, CNS, and systemic circulation. The intestinal neural plexus is exposed directly to oxidative stress, pro-inflammatory cytokines, toxins, and pathogenic bacteria. Reduced intestinal barrier integrity is linked with reduced BBB integrity and neuroinflammation.
Inflammation	*Bacteroides* ↑ *Akkermansia* ↑ *Oscillospira* ↑ *Ralstonia* ↑	Increases in pro-inflammatory microbes trigger an increase in pro-inflammatory cytokines through enhanced levels of lipopolysaccharides (LPS) within the GIT.	Excess LPS levels trigger neuroinflammation, neuronal death, and the release of α-synuclein, culminating in progressive SN dopaminergic neurodegeneration and decreased striatal dopamine levels. LPS induces nitrosative and oxidative stress, and activates microglia, causing subsequent production of pro-inflammatory cytokines, worsening GIT inflammation, increasing intestinal permeability, and disrupting the BBB [8,14,37].
	*Bifidobacterium* ↓	*Bifidobacterium* prevent the overgrowth of harmful bacteria and regulate the GI immune system.	Low *Bifidobacterium* counts have been associated with cognitive symptoms, including worsening of hallucinations.
Oxidative stress	SCFA-producing microbes ↓ *Escherichia coli* ↑ *Salmonella enterica* ↑	The loss of SCFA-producing microbes increases the partial oxygen pressure in the GIT, which allows oxygen and nitrates to diffuse into the lumen. Oxidation of the GIT allows the proliferation of oxygen-tolerant microbes, such as pathogenic *Escherichia coli* or *Salmonella enterica* species [41,44].	Oxidative stress leads to carbonylation of proteins and protein aggregation, a key pathology in the nigrostriatal dopaminergic neurons of PD patients’ brains that leads to neuroinflammation [15]. Oxidative stress leads to the activation of various inflammatory cascades and activation of enteric neurons and glial cells in the ENS, CNS, and vagus nerve, which contributes to accumulation and misfolding of α-synuclein in the ENS, which can transfer to the brain. In the brain, pathologic α-synuclein, pro-inflammatory cytokines, and immune cells, together with tissue debris or abnormal proteins released from lysed cells, cause microglial activation and neuroinflammation, leading to the dysfunction and degeneration of dopaminergic neurons in the SNc [1,14,16,19].
Pathogenic bacteria overgrowth	*Heliobacter pylori* ↑	In some individuals, *H. pylori* can attach to gastric epithelial cells and damage the lining of the stomach and small intestine by releasing proteins, such as vacuolating cytotoxin A. *H. pylori* also contains LPS, and these endotoxins can promote inflammation and apoptosis in the gut, as well as increasing inflammatory cytokines.	PD patients with *H. pylori* experience more severe motor symptoms, more rapid disease progression, and decreased dopamine in the brain. A recent meta-analysis found significant associations between *H. pylori* infection and mean unified Parkinson’s disease rating scale (UPDRS) scores [48].
	*Streptococcus* ↑	Can produce neurotoxins, such as streptomycin, streptodornase, and streptokinase.	These toxins can lead to permanent neurological damage.
	*Enterococcus* ↑	Produces extracellular superoxide and hydrogen peroxide.	*Enterococcus* damages colonic epithelial cell DNA.
	*Escherichia-Shigella* ↑	Causes diarrhoea and produces Shiga toxin.	*Escherichia-Shigella* may cause functional lesions in the central nervous system.
Small intestinal bacterial overgrowth (SIBO)	*Enterobacteriaceae* (*family*) ↑ (increased colonisation within the small intestine)	SIBO can cause symptoms including diarrhoea, flatulence, and abdominal pain and swelling. SIBO may also be implicated in ‘leaky gut’, as it can increase permeability via the local inflammatory response generated.	SIBO triggers interactions between α-synuclein, endotoxins, and other molecules, which can cross the intestinal barrier and trigger microglial activation, prompting α-synuclein nucleation. SIBO is present in 25% of all PD sufferers, and up to 55% of those with fluctuating motor symptoms [9,37].
Metabolome depletion	*Lachnospiraceae* (*family*) ↓ *Blautia* ↓ *Coprococcus* ↓ *Roseburia* ↓	A reduction in metabolite production, specifically SCFAs, by commensal bacteria leads to increased GI, systemic, and neurological inflammation, increased intestinal and BBB permeability, and changes in endocrine signalling.	Reduction in SCFAs increases susceptibility to ENS infection of enteric pathogens and increases the risk of α-synuclein formation in the ENS. A study of 111 subjects confirmed that SCFAs are decreased in PD patients’ faecal samples, where faecal levels of butyric acid and inflammatory markers (IL-8 and IL-1β) correlated with age at disease onset [49]. Another study also found significant absolute and relative reductions in faecal SCFAs in PD patients, which was consistent with the observed alterations in gut microbiota composition [29].
	*Prevotellaceae* (*family*) ↓ *Faecalibacterium* ↓ *Ruminococcus* ↓	The *Prevotellaceae* family have a role in the biosynthesis of essential vitamins, such as thiamine and folate, and niacin (vitamin B3) biosynthesis.	Reduced biosynthesis of thiamine and folate is linked with worsening neurodegeneration, motor function, and tremors, while also impacting dopamine release. Niacin is a precursor of nicotinamide adenine dinucleotide (NAD), which has been found to be reduced in PD patients’ plasma samples.
Protein misfolding	Amyloid-protein-producing Gram-negative bacteria ↑ e.g., *Streptococcus mutans* ↑ *Staphylococcus aureus* ↑ *Escherichia coli* ↑	An increase in amyloid-protein-producing bacteria leads to an increase in Curli protein formation. Curli proteins form biofilms in gastro-epithelial tissue, they can evade the human immune response, and lead to constipation and irritable bowel disease (IBD) symptoms.	Curli proteins interact with human α-synuclein, inducing nucleation and aggregation. The α-synuclein then furthers this reaction in a prion-like manner, by triggering the misfolding of neighbouring α-synuclein proteins. When α-synuclein deposits into dopaminergic neurons in the brain, it results in motor symptoms [18,23,37]. Oral administration of Curli-producing *E. coli* also resulted in increased pro-inflammatory cytokines (TLR2, TNF, and IL-6) and neuroinflammation in several brain regions, such as the striatum, SN, and the hippocampus [8,10].
Changes in neurotransmitter signalling	*Bacteroides* ↑ *Parabacteroides* ↑ *Escherichia* ↑	These species actively suppress the GABA production pathway in stool and produce GABA in the intestines.	GABA is transferred to the brain from the intestines via the gut–brain axis. Increased GABA in the pons, basal ganglia, and thalamus is associated with the degree of bradykinesia and rigidity in PD patients.
	*Lactobacillaceae* (*family*) ↓ *Prevotellaceae* (*family*) ↓	Reductions in ghrelin-secreting bacteria alter ghrelin levels, which alters GI dopamine levels. Changes in GI dopamine levels impact gut motility [18,50].	Ghrelin is an integral GI hormone involved in dopamine function of the substantia nigra and striatum via alterations in mitochondrial respiration and reactive oxygen species (ROS) production. Changes in ghrelin secretion trigger changes in dopamine levels in the GI, which influences dopamine metabolism via receptors in the ENS. Ghrelin also plays a role in protecting dopaminergic neurons by preventing a-synuclein accumulation and phosphorylation, as well as promoting autophagy and inhibiting apoptosis [8,22].

* Other compositional changes in the microbiome (or forms of gut dysbiosis) are also responsible for triggering the defined mechanisms. The examples provided are known microbial shifts, but the examples provided are not extensive. ↑ = increase in abundance; ↓ = decrease in abundance.

**Table 3 biomedicines-12-01738-t003:** A summary of antibiotic agents that have been employed to exert neuroprotective effects or alleviate Parkinson’s Disease symptoms through defined mechanisms of action. Adapted from [16,20,21].

Antibiotic	Bacterial Target	Mechanism of Action
Ampicillin	Group A Streptococcus (responsible for the impairment of the central dopaminergic system).	Ampicillin exerts its neuroprotective effect via modulating the gut–brain axis in PD.
Ceftriaxone	Broad spectrum	Ceftriaxone (CTX) is a broad-spectrum tetracycline, β-lactam antibiotic that can cross the BBB and is known to increase glutamate transporter subtype 1 (GLT-1) expression; thus, it can increase glutamate uptake and reduce excitotoxicity. It binds specifically to α-synuclein and inhibits its oligomerisation process. Reported to have antioxidant and anti-inflammatory effects.
Doxycycline	Broad spectrum	Doxycycline is a broad-spectrum antibiotic with anti-apoptotic and anti-inflammatory mechanisms.
Minocycline	Broad spectrum	Minocycline can cross the BBB and has anti-inflammatory, anti-apoptotic, and neuroprotective effects.
Neamine	Broad spectrum	Aminoglycoside antibiotic can reduce BBB leakage and apoptosis.
Rifampicin	Broad spectrum	Rifampicin is a semi-synthetic macrocyclic antibiotic that tightly binds to α-synuclein and inhibits its fibrillation process, preventing the formation of α-synuclein aggregates, as well as disaggregating existing pre-formed fibrils.

**Table 4 biomedicines-12-01738-t004:** A summary of faecal microbiota transplantation (FMT) clinical study designs and outcomes for the treatment of Parkinson’s Disease [2,36,40,42,67,69,70].

Reference	Study Design	Administration Route, Formulation, and Frequency	Eligibility Criteria	Donor Information	Microbiome Analyses	Clinical Outcomes	Adverse Effects
Bruggeman et al. 2024 [69]	Randomised, double-blind, placebo-controlled trial (N = 46, 22 = donor FMT (15M, 7F)/24 = placebo FMT (14M, 10F), FMT with own stool). It is noted that the final results include 43 participants who completed all study visits (21 in the donor FMT group and 22 in the placebo FMT group).	50 g of faecal product was diluted with sterile saline and then homogenised anaerobically and filtered. Glycerol (10%) was added as a cryoprotectant as samples were stored at −80 °C until use. This resulted in a total volume of 200 mL. The suspension was thawed and administered into the intestine via the naso-jejunal tube. After treatment, participants were followed up at 3, 6, and 12 months after FMT.	Patients aged 50–65 years, with a clinical PD diagnosis according to the Movement Disorder Society (MDS) criteria, Hoehn and Yahr stage 2 or 3 in an off-medication state, and onset of motor symptoms onset after age 50. Patients were excluded if they had a first-degree relative/more than one relative with PD, a diagnosis of dementia or Mini-Mental State Examination Score < 25, a diagnosis of depression or psychosis (DSM-V criteria), GI dysfunction unrelated to PD, an immune disorder, or were under clinical immunosuppression. Drug abuse, malignancy, or any severe comorbidity that might interfere with the study course were also considered exclusion criteria, as was the use of probiotics or antibiotics in the three months prior to the FMT, or respiratory tract infection in the two months prior to the FMT.	Healthy donors were recruited via the Ghent Stool Bank following a strict inclusion protocol and selection process, which included a review of the donor’s clinical and personal information as well as serology and stool testing. Stools that were collected before the COVID-19 pandemic were used.	No analysis of the microbiome profile before or after the FMT.	There were improvements in MDS-UPDRS part 3 (off-medication) scores in both the donor FMT (decrease of 5.8 points) and placebo FMT groups (decrease of 2.7 points). The change in MDS UPDRS motor score from baseline to 12 months post-FMT was significantly different between treatment groups. The donor FMT group also exhibited a slight decrease in GI transit time, and worse performance on the Parkinson’s Fatigue Scale. There were no significant differences between the groups in other scores of the MDS-UPDRS, the levodopa-equivalent daily dose (LEDD), the Non-Motor Symptoms Scale for Parkinson’s Disease, the Parkinson’s Disease Quality of Life Questionnaire, Wexner Constipation Scale, Geriatric Depression Scale, Parkinson Anxiety Scale, Lille Apathy Rating Scale, Parkinson’s Disease Sleep Scale, and Montreal Cognitive Assessment.	There were no severe adverse events (AEs) associated with treatment or placebo. Mild transient GI AEs (abdominal cramps and nausea) were reported in the first week after treatment: 13 (59%) patients in the donor FMT group and 6 (25%) patients in the placebo group.
Cheng et al., 2023 [70]	Placebo-controlled randomised clinical trial for (N = 54, 27 = FMT (15M, 12F)/27 = control (17M, 10F)).	Oral administration: 16 capsules (FMT or placebo) were taken orally by participants in the morning on an empty stomach. Capsules were taken once a week for 3 consecutive weeks. After treatment, patients were followed up in the 4th, 8th, and 12th weeks.	Early PD diagnosed as having Hoehn and Yahr scale (HY) 1~3 grade of PD in accordance with the diagnostic criteria for PD in China, with no specific difficulty in communication.	4 stool donors were selected according to rigorous health screening. Each donor provided stools for making FMT capsules for about 7 patients, and each patient in the FMT group was given 16 FMT capsules at each time, which are made from approximately 50 g of donated stool. Donor–recipient pairing was not performed in this study, and all patients received FMT from a random donor each time.	16S rRNA sequencing for stool samples from donors (3 times during stool donation), as well as patients at baseline and the 4th and 12th weeks after FMT or placebo intervention.	FMT participants demonstrated a significantly greater change in MDS-UPDRS total score: PD symptoms were alleviated. Non-motor experiences in the daily life of patients (MDS-UPDRS part 1 scale) were significantly improved by FMT, at week 4 and week 12. Non-motor aspects not only included improved GI symptoms but also cognitive function. FMT treatment participants showed better outcomes in cognitive function through the Montreal Cognitive Assessment (MoCA). High placebo response observed with GI symptoms at weeks 4 and 8. This may be due to dietary recommendations given to both FMT and control subjects. However, by the end of the trial, FMT subjects’ improvements were greater. This study also categorised FMT subjects into FMT responders (*n* = 13) and non-responders (*n* = 14). FMT responders showed obvious improvement in PD symptoms after treatment, while others did not (according to the change in the MDS-UPDRS Part 2 score).	6 minor adverse events (AEs; bloating, flatulence, nausea, diarrhoea); however, it is noted that 3 were in FMT subjects and 3 in the placebo group.
DuPont et al., 2023 [67]	Randomised, double-blind, placebo-controlled pilot study (N = 12; however, one subject withdrew before completion due to unrelated diagnosis of cancer). Subjects were followed for safety and clinical improvement for 9 additional months (total study duration 12 months).	Orally administered lyophilised FMT product (60 g of donor faeces, or placebo) was given twice weekly for 12 weeks (24 doses).	Mild to moderate PD with constipation. Enrolment criteria included PD diagnosis by UK Brain Bank criteria, robust response to dopaminergic therapy defined as ≥33% reduction of the UPDRS motor score in the OFF vs. ON-dopaminergic medication state, and a MoCA of >23. Mild/moderate PD was determined by ≤10 years of disease duration from the date of initial diagnosis, an OFF-medicine-state, HY of ≤3, and the absence of certain non-motor symptoms, including dementia, postural instability, and dysphagia. Constipation was considered hard/difficult to pass stools with no more than 3 bowel movements per week.	Four thoroughly screened donors provided all FMT products. Two capsules from each of three donors (a total of six capsules) were combined for treatment of each of the first seven subjects randomised to FMT. For the eighth subject, on request of the FDA, the last two FMT doses, capsules from one of two donors, were given sequentially for the remaining two treatments.	Stool specimens were provided by study subjects for microbiome examination before treatment, at weeks 6 and 13 (1 week after treatment), and at 4, 6, and 9 months after completing treatment, and stored at −80 °C until sequencing. Whole Metagenome Shotgun sequencing of stool samples was conducted at the Baylor College of Medicine Center for Metagenomics and Microbiome Research.	Beta diversity (taxa) of the microbiome increased significantly at 6 weeks (*p* = 0.008) and 13 weeks (*p* = 0.0008) for subjects randomised to FMT (diversity was similar in placebo and FMT groups at baseline). After FMT, proportions of selective families within the phylum Firmicutes increased significantly, while proportion of microbiota belonging to Proteobacteria were significantly reduced. Objective motor findings showed only temporary improvement, while subjective symptom improvements were reported compared to baseline in the group receiving FMT. Constipation, gut transient times (NS), and gut motility index (*p* = 0.0374) were improved in the FMT group.	Adverse events were reported in 7 (88%) FMT and 4 (100%) in placebo-treated subjects (bloating/flatulence 2–0, abdominal pain/cramps/discomfort 3–1, worsening constipation 1–2, diarrhoea 2–0, nausea 1–0, and pre-existent gall stone symptoms 1–0). GI complaints occurred more commonly in the active treatment group and were determined to be probably related to FMT. All adverse events were transient and mild (47%) or moderate (38%) in severity. No AEs were persistent, and no treatments were withheld because of them. Safety laboratories did not identify clinical abnormalities.
Kuai et al., 2021. [40]	Prospective study (N11, 7M, 4F).	40–50 mL of frozen faecal microbiota was suspended in 200 mL of warm normal saline, and transplanted into the intestine, within 2–4 min of the suspension, through a nasoduodenal tube.	PD with constipation. Patients were excluded if they had severe immunodeficiency, obvious liver and kidney dysfunction, could not provide informed consent, or were accompanied by *C. difcile* infections or other intestinal pathogens.	Frozen faecal microbiota was obtained from the China fmtBank (Nanjing, China).	Faecal samples from all patients before FMT and 4, 8, and 12 weeks after FMT, and those of healthy controls, were collected and stored at −80 °C until usage. The 16s rDNA sequencing was used for the microbiota analysis. The phylum- and family-level analyses were used to assess the composition of the faecal bacteria. In addition, the Shannon diversity index and chao1 index were used to assess the microbiota diversity. Lactulose H2 Breath Testing was conducted to identify SIBO, one week before FMT treatment.	FMT was an efficient treatment for constipation and other GI symptoms in PD patients, with symptoms improved for at least 12 weeks. Other clinical metrics measured (H-Y Grade, UPDRS II Score, NMSS, and PAC-QoL) also showed improvement. SIBO was detected in all PD patients at baseline. This was corrected by week 12 after FMT. At baseline, microbiota species diversity and the pattern of the richness of PD patients’ samples were significantly decreased compared to HCs (*p* < 0.05). Richness and diversity indices in PD patients 12 weeks post-FMT were significantly increased to levels that were not significantly different from HCs.	During treatment, the most common adverse events were mild diarrhoea (9.1%), abdominal pain (27.3%), venting (18.2%), flatulence (45.5%), nausea (27.3%), and throat irritation (18.2%). All cases were mild, and none resulted in the discontinuation of the treatment. During follow-up, abdominal pain (18.2%) and flatulence (18.2%) were also the most common adverse events. Other adverse events were venting (9.1%) but were self-limiting and not serious. No serious adverse effects occurred during treatment or follow up.
Segal et al., 2021 [42]	(N = 6, 3M, 3F) Case series: clinical outcomes assessing motor, non-motor, and constipation symptoms were compared at baseline and at 2, 4, 8, 12, 16, 20, and 24 weeks after the FMT.	Colonoscopy: 300 mL of faecal suspension of donor stool was delivered in three portions—100 mL at the terminal ileum, 100 mL at the cecum, and 100 mL along the rest of the colon. Prior to the colonoscopy (the day before), participants received 4 L of macrogol bowel preparation. To avoid additional changes in the gut microbiome, participants were asked not to make changes in their diets, and no antibiotics were used throughout the study period.	Mild to moderate PD with constipation. Patients also needed to meet criteria for screening colonoscopy of colorectal cancer (age over 50 or a history that warrants colonoscopy).	Stool donors underwent rigorous screening and physical examination. Use of antibiotics in the 6 months preceding the donation was not permitted. Two healthy donors, males aged 38 and 50 years, were recruited. Donation transplanted in each PD patient was double-blinded on the day of the procedure and unblinded at the time of data analysis.	No analysis of the microbiome profile before or after the FMT.	FMT was safe and resulted in improvements in PD motor and non-motor symptoms, including constipation, at 6 months. Patients with longer disease duration and worse initial symptoms had a more significant improvement compared to those with mild disease. Four weeks following the FMT, motor, non-motor, and constipation scores were improved in 5 of 6 patients. At week 24, compared to baseline, the changes in motor scores ranged from −13 to 7 points, in non-motor scores from −2 to −45 points, and in constipation scores from −12 to 1 point.	One patient had a serious AE requiring admission for observation only (recurrent episodes of vasovagal pre-syncope that appeared 24 h after the FMT and lasted for 8 h). No AEs were observed in all other patients.
Xue et al., 2020 [36]	(N = 15, 11M, 4F)	10 received colonic FMT and 5 received naso-intestinal FMT. Faecal matter (from donors) was purified and isolated. Single faeces from a donor were used to isolate microbiota to treat a PD patient. The time between faeces being released and being transplanted into the gut of a PD patient was less than one hour. All patients continued their previous medication before FMT and the following 3 months after FMT.	PD patients failed to achieve satisfactory effectiveness from the previous medications.	Healthy donors (ranged from 18 to 24 years old) underwent rigorous screening and biomedical testing. Five Chinese Han donors with a mean age of 22 years, including 3 men and 2 women, participated in this study. The selection of the donor was random.	No analysis of microbiota composition or blood inflammatory factors.	Three patients refused to continue follow-up because of unsatisfactory efficacy after 1 month of FMT treatment. Therefore, all patients (15 patients with PD) completed 1-month follow-up, but only 12 patients completed the 3-month follow-up after FMT. The mean score of UPDRS-III decreased from 41.75 before FMT to 24.00 at 3 months after FMT. The quality of sleep and life improved after FMT treatment, with a decrease in PSQI score (12.41 vs. 8.16) and PDQ-39 score (52.16 vs. 25.91). In addition, anxiety and depression were also partially relieved, with a significant decrease in HAMA score (21.08 vs. 9.58) and HAMD score (22.41 vs. 10.08). Colonic FMT showed better results than nasal intestinal FMT: Two patients achieved self-satisfying outcomes that last for more than 24 months. Conversely, the effect of FMT via the naso-intestinal tube was unsatisfactory, especially for non-motor symptoms.	5 patients experienced AEs (3 in colonic group, 2 in naso-intestinal group). All adverse events were mild (abdominal pain, diarrhoea, and flatulence).
Huang et al., 2019 [2]	Case study (N = 1, M).	Colonoscopy: 200 mL of prepared faecal microbiota suspension was injected through the TET tube, 3 times over 3 days.	71-year-old male patient presented with 7 years of resting tremor and bradykinesia that first inflicted the upper limbs and subsequently spread to the right lower limb and left lower limb. Additionally, patient experienced constipation (defined as defecation needing more than 30 min) for years. UPDRS scores were given as follows: part II (difficulties in activities of daily living), 13/42; part III (motor examination), 46/142. Patient-assessment of constipation quality of life (PAC-QOL score and Wexner constipation score, 18 and 16).	The stool for FMT was obtained from a college student (26-year-old male) who met screening criteria.	Stool sampled for 16s RNA microbiota analysis, PCoA analysis, OTUs clustering, and weighted unifrac tree analysis method.	Tremor in legs almost disappeared at 1 week after FMT treatment. Resting tremor recurred in the right lower extremity at 2 months after FMT, however at reduced severity compared with that of pre-FMT treatment. The patient’s UPDRS score began to decrease after FMT. This decrease became significant at 1 week after treatment, but later, the score showed a trend of increasing with time. PAC-QOL and Wexner constipation scores suggested that the patient’s constipation was significantly alleviated. Together with the results of faecal microbiota analyses at different follow-up points, this study showed that FMT could effectively raise the alpha diversity of the gut microbiota.	No adverse reaction appeared during FMT.

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
