# Peer review of "The Gut Microbiome as a Catalyst and Emerging Therapeutic Target for Parkinson’s Disease: A Comprehensive Update"

_biomedicines, 2024, doi:10.3390/biomedicines12081738_

Round 1

Reviewer 1 Report

Comments and Suggestions for Authors

The reviewer would like to declare no conflict of interest with the authors and their affiliations.

The authors presented an interesting review topic on the possible contribution of gut dysbiosis to Parkinson's disease. However, some minor comments need to be addressed before the manuscript can be accepted for publications.

1. The current review skewed heavily towards the "gut-to-brain" transmission of alpha-synuclein/Lewy body as the main cause of PD. The gut-to-brain transmission hypothesis of PD is relatively new idea and it is not a commonly known causative action for PD. Besides direct vagal transmission, gut-brain-axis also involve other means of communications eg. hormonal, humoral, immune signalings etc. The authors should also equally discuss these mode of signalling.

2. Another notion of "direct colonic adnimistration" (line 190) should be removed. This "route" of administration is very uncomoon and very unlikely to be done in humans. 

3. Another rather assertive/over-claimed note is observed in line 232, microbio-targeted therapies are very far-fetched to be considered as "first-line pharmacotherapy" for PD. It is consistently observed in the manuscript that the authors tend of "over-exaggerate" or too assertive on the description, the authors should "tone down" on the claims and suggestions.

Author Response

Comment 1: The reviewer would like to declare no conflict of interest with the authors and their affiliations.

The authors presented an interesting review topic on the possible contribution of gut dysbiosis to Parkinson's disease. However, some minor comments need to be addressed before the manuscript can be accepted for publications.

Response 1: The authors would like to thank the Reviewer for their thorough assessment, positive feedback, and constructive comments.  

Comment 2: The current review skewed heavily towards the "gut-to-brain" transmission of alpha-synuclein/Lewy body as the main cause of PD. The gut-to-brain transmission hypothesis of PD is relatively new idea and it is not a commonly known causative action for PD. Besides direct vagal transmission, gut-brain-axis also involve other means of communications eg. hormonal, humoral, immune signalings etc. The authors should also equally discuss these mode of signalling.

Response 2: The authors would like to thank the Reviewer for their comment. The authors agree that there are many pathways of gut-to-brain communication which could be involved in PD etiology, however it is beyond the scope of the paper to outline all mechanisms in detail. Given these mechanisms have been previously reviewed in detail, the text has been edited to direct the reader to these comprehensive reviews to ensure complete pathways are covered in this manuscript.

Comment 3: Another notion of "direct colonic adnimistration" (line 190) should be removed. This "route" of administration is very uncomoon and very unlikely to be done in humans. 

Response 3: The authors would like to thank the Reviewer for highlighting the lack of feasibility and translatability of ‘direct colonic administration’. This section of the text has been subsequently removed.

Comment 4: Another rather assertive/over-claimed note is observed in line 232, microbio-targeted therapies are very far-fetched to be considered as "first-line pharmacotherapy" for PD. It is consistently observed in the manuscript that the authors tend of "over-exaggerate" or too assertive on the description, the authors should "tone down" on the claims and suggestions.

Response 4: The authors would like to thank the Reviewer for their constructive criticism. We agree that utilising microbiome-targeted therapies as frontline pharmacotherapy may serve as an overly ambitious statement without the current clinical evidence to support this. Therefore, the authors have “toned down” this language within the section highlighted by the Reviewer, and throughout the entire manuscript, to ensure that the potential and capacity for gut-microbiome targeted therapies to treat PD is not over-exaggerated. 

Reviewer 2 Report

Comments and Suggestions for Authors

This is a comprehensive review summarizing the state of the art on the role of gut microbiome in the pathophysiology of Parkinson's disease. Also, promising innovative therapeutic strategies, including antibiotics and probiotics as well as faecal microbiota transplantation have been critically discussed. The review is well-designed, organized, and full of detailed information on specific microbiological issues of gastrointestinal tract microbiota. I have only some minor comments.

First, it is unclear whether suggested treatments are mutually exclusive or can be combined to improve efficacy. Also, it is unclear whether the therapeutic strategies can be used also in the early stages of the disease or by contrast they should be adopted early on the disease course (for instance in prodromal stages).

The role of genetic advances in the microbiota characterization should be expanded. 

Comments on the Quality of English Language

English editing is required throughout the paper to control for typos and spelling errors.  

Author Response

Comment 1: This is a comprehensive review summarizing the state of the art on the role of gut microbiome in the pathophysiology of Parkinson's disease. Also, promising innovative therapeutic strategies, including antibiotics and probiotics as well as faecal microbiota transplantation have been critically discussed. The review is well-designed, organized, and full of detailed information on specific microbiological issues of gastrointestinal tract microbiota. I have only some minor comments.

Response 1: The authors would like to thank the Reviewer for their thorough assessment, positive feedback, and constructive comments.

Comment 2: First, it is unclear whether suggested treatments are mutually exclusive or can be combined to improve efficacy. Also, it is unclear whether the therapeutic strategies can be used also in the early stages of the disease or by contrast they should be adopted early on the disease course (for instance in prodromal stages).

Response 2: The authors would like to thank the Reviewer for their comment. We must emphasise that both strategies have been tested, i.e. microbiome-targeted therapies have been investigated as standalone and adjuvant therapies to existing treatment options. Where there is evidence to show combination treatments it has been included (such as synbiotics, or dietary measures to support FMT). Unfortunately, as these treatments are emerging there is little literature available. Likewise for which stage of disease is best to target. Some text has been added to the conclusion to highlight that these areas are yet unknown. Additionally, throughout the text, it has been mentioned that there is need for more research in this space and thus these strategies have been highlighted as potential adjunct therapies. 

Comment 3: The role of genetic advances in the microbiota characterization should be expanded.

Response 3: Thank you to the reviewer for raising this. Further text has been added to this section of the manuscript to highlight the importance of advances in this space.